# Image-Based Concrete Crack Detection Method Using the Median Absolute Deviation

**DOI:** 10.3390/s24092736

**Published:** 2024-04-25

**Authors:** Juan Camilo Avendaño, John Leander, Raid Karoumi

**Affiliations:** Division of Structural Engineering and Bridges, KTH Royal Institute of Technology, 10044 Stockholm, Sweden; john.leander@byv.kth.se (J.L.); raid.karoumi@byv.kth.se (R.K.)

**Keywords:** crack detection, probability of detection, median absolute value, thresholding, computer vision, damage detection

## Abstract

This paper proposes an innovative approach for detecting and quantifying concrete cracks using an adaptive threshold method based on Median Absolute Deviation (MAD) in images. The technique applies limited pre-processing steps and then dynamically determines a threshold adapted for each sub-image depending on the greyscale distribution of the pixels, resulting in tailored crack segmentation. The edges of the crack are obtained using the Laplace edge detection method, and the width of the crack is obtained for each centreline point. The method’s performance is measured using the Probability of Detection (POD) curves as a function of the actual crack size, revealing remarkable capabilities. It was found that the proposed method could detect cracks as narrow as 0.1 mm, with a probability of 94% and 100% for cracks with larger widths. It was also found that the method has higher accuracy, precision, and F2 score values than the Otsu and Niblack methods.

## 1. Introduction

Bridges are crucial assets connecting regions and enabling transportation, making them fundamental for a society’s development [1]. Assuring their correct functioning requires recurring inspections of the structures at regular intervals, in order to detect possible damage [2]. Damage detection is important in infrastructure management since it helps the assessment of deterioration over time due to different causes, such as corrosion, creep, and cyclic loading, among others. These procedures aim to detect damages in bridges and concrete structures subjected to different adverse conditions that can lead to their deterioration. These hazards can come from natural processes such as earthquakes, flooding, landslides, and human construction errors. Another cause for damage is the effect of loads on the structure (wind load, vehicle load, thermal loads, and others), which can produce damages. Among the most common types of damage found are cracks, which serve as early signs of deterioration and are, in many cases, made by flexion or shear forces as well as temperature-induced stresses. Cracks are therefore important to study to ensure structural reliability and integrity of structures. Thus, efficient management and maintenance operations, based on identified damage, can considerably extend a structure’s life span [3].

For civil infrastructure, the basic and main method used for assessing condition, both functional and physical, is visual inspection [4]. However, this kind of inspection has some limitations because it involves the use of heavy machinery, elevated costs, and risky situations for the personnel performing the work, and the results can fluctuate between different inspectors due to human error [5,6,7,8].

To address these limitations, methods of Structural Health Monitoring (SHM) have been developed and used in various fields, such as mechanical, aerospace, and civil engineering, to ensure the safety and integrity of assets. SHM involves strategies to detect damage using various methods (hardware or software) and obtain data that can be analysed to properly manage and maintain the assets [9]. Within SHM, there are several methods and they can be classified as either contact or non-contact techniques. Contact sensing methods involve sensors attached to a structure to measure accelerations, displacements, and inclinations. These can pose a constraint from an economic point of view, as well as generate practical challenges, in terms of the time and labour cost of the installation process [10].

Non-contact methods have gained popularity due to the economic advantages and practical benefits they present. Methods using cameras have been developed for image-based inspections [10]. These methods are becoming an important part of SHM for evaluating different types of objects, their location, and understanding their context. The process of analysing the images and their content is called ‘computer vision’ and can be used to detect damage [11]. Some processing techniques applied to images are: histogram transforms, background subtraction, texture recognition, filtering, or edge detection, and others [4]. Filtering is one of the most common operations used in computer vision and can be sorted into two different categories: space domain algorithms and frequency domain algorithms [12]. Space domain filtering focuses on pixels directly from the image. On the other hand, frequency domain filtering uses transformations on the pixels (using Fourier transform or Wavelet transforms) and then transforms them back to the space domain. Commonly used space domain filters include the median filter, mean filter, Gaussian filter, and bilateral filtering [13].

One of the most common processing techniques used to extract crack information is the ‘edge detection algorithm’. Crack edge detection algorithms have been developed by adapting methods used in other disciplines; e.g., for text recognition [14], the Sauvola method has been used to identify text from images with a noisy background [15]. Researchers have used this knowledge and applied it to civil engineering structures to delimit the crack boundaries and differentiate pixels from cracks and the background [16,17,18]. Once the pixels are delimited, the background and crack pixels are usually transformed into black and white, with the help of image binarisation using threshold methods, such as the Niblack, Otsu, or Wolf methods [8,19].

Based on these procedures, researchers have developed algorithms like Crack Width Transform to determine a crack candidate region and then filter by using a threshold [20]. Kim et al. [19] used a median filter to remove noise and Sauvola’s binarisation method to transform the image from greyscale to black and white, identifying cracks. Later, Kim et al. [15] used the Sauvola method with two different sets of parameters to detect cracks and determine their widths and lengths.

Other methods use histogram-based algorithms to study cracks in concrete or pavements [21]. Kapela et al. [22] used Histograms of Oriented Gradients (HoG) for crack detection by computing the intersection of Gaussian functions in intensity histograms to determine a segmentation threshold. Li et al. [23] proposed a threshold algorithm based on the Neighbouring Difference Histogram Method (NDHM) that uses the standard deviation of an image to obtain the result. Histogram-based thresholding methods are popular due to their simplicity and high efficiency, but using a general threshold makes the results susceptible to noise [24]. Although image thresholding can be used for identifying cracks, and it is among the most popular methods [25], the results are highly dependent on the threshold parameters, which cannot be used for every situation [15].

In recent years, the development of Unmanned Aerial Vehicles (UAVs) has led to cost-efficient data collection when accessing difficult locations to perform image-based inspections [26]. In [27], Dorafshan et al. compared different processing algorithms for UAV-assisted crack detection, such as filtering, edge detection, image enhancement, and segmentation. The edge detection was carried out using six filters: Roberts, Prewitt, Sobel, and Laplacian of Gaussian (in the spatial domain) and Butterworth and Gaussian (in the frequency domain). The minimum detectable crack width was found to be 0.2 mm. They concluded that the Laplacian of Gaussian filter provided the fastest and most accurate method for studying crack width.

Machine learning-assisted crack detection is another approach to image-based inspection that has profited from the development of computational power. This approach is influenced by image processing methods, which are used as the first step to extract damage features from the images before the machine learning algorithm is trained [28]. Particularly, Convolutional Neural Networks (CNNs) are one of the most prominent methods used to evaluate damages in images. This method falls into the category of supervised learning since the models are trained for specific tasks such as detecting cracks or corrosion in structures using datasets from which they learn relevant features for damage classification. Although effective, machine learning algorithms have challenges and limitations. One significant challenge is the reliance on private datasets, i.e., datasets created by the researchers using the model and not public datasets that can be easily accessed [29]. Models trained on such datasets often achieve high levels of accuracy but perform poorly when exposed to datasets with different characteristics, such as varying lighting conditions, surface textures, and noisy surroundings [30]. Additionally, the creation of these datasets needs meticulous labelling efforts, particularly for pixel-level segmentation, which is highly time-consuming. Moreover, these models tend to be more complex structures than computer vision algorithms, with higher inference times and the need for manual input of the training parameters [2]. Furthermore, the computational demand of these models often requires significant computational hardware, such as Graphic Processing Units (GPUs), which may not be available to all researchers [3,12,29].

The inspection strategies presented require understanding the types and characteristics of the defects they can reliably detect [31]. Nevertheless, a small amount of research has been conducted to determine the accuracy and reliability of vision-based inspections. For this, the Probability of Detection (POD) is used to determine a method’s capability metric, and the probability of detecting a certain type of damage using a specific method [32].

As previously discussed, several different algorithms exist for detecting and analysing cracks in images, each involving different pre-processing filters, such as mean filter, median filter, or Gaussian filter. Similarly, various thresholding techniques, such as the Otsu method, Sauvola method, and histogram-based methods, are available. Additionally, for edge detection, we have filters, such as Roberts, Prewitt, Sobel, and more, to delimit the boundaries of a crack. Different combinations of these methods may lead to different outcomes; some filters, such as the Gaussian filter for example, might blur the edges, reducing the reliability of the edge detection. Determining the appropriate combination of methods for pre-processing, segmentation, and quantification is not a straightforward task.

In this paper, we propose a novel approach to histogram-based thresholding using the Median Absolute Deviation (MAD) within subsections of images. This approach offers advantages over other methods, including eliminating labelling datasets as in the machine learning approach, reducing the amount of pre-processing techniques, and simplifying the overall procedure compared to algorithms combining multiple methods. The use of MAD enables the establishment of an adaptable threshold for each subsection, eliminating the general threshold. By doing so, we eliminate the possibility of a single threshold value being suitable for some images but not for others. Additionally, the paper presents POD curves to assess the method’s capability to detect cracks of varying sizes. The image processing focuses on the pixel intensities to calculate the MAD and determines a threshold that can separate crack pixels from the background. Subsequently, we employ the Laplacian edge detector to delimit the crack. The widths of the cracks are obtained, and the results are used to determine the method’s accuracy.

## 2. Methodology

The algorithm employed in this paper consists of three different main stages: crack detection within the image, pixel extraction, and crack width measurement.

The images obtained were pre-processed by dividing them into sub-images and converting them to greyscale, in order to simplify the study of the pixel intensities. To differentiate background and crack pixels, a specific threshold value was determined for each sub-image, using the Median Absolute Deviation (MAD). Subsequently, a thresholding procedure was applied to isolate the potential crack region. Edge detection was employed in the crack regions, to delimitate the boundaries of the detected crack. This was performed using the Laplacian function. After this step, the centreline of the crack was obtained through the ‘skeletonize’ function. For each pixel that is part of the skeleton centreline, the distances to the nearest edges were calculated. The distance measured corresponded to the width of the crack at the specific point.

### 2.1. Data Acquisition

The data acquisition employed in this study aimed to gather images containing concrete cracks that included widths between 0.1 and 1.5 mm. The imaging was performed with a Nikon D810 digital camera, coupled with an AF-s Nikon 24–70 mm f/2.8 G ED lens. The sensor in the digital camera had a resolution of 36 megapixels. The images were taken with the highest available resolution provided by the camera, obtaining a resolution of 7380 × 4928 pixels per image. For optimal contrast, the ISO used was 125. The camera was mounted on a tripod at a distance of approximately 1 m from the sensor to the concrete surface, in order to obtain a pixel size of 0.1 mm and avoid blurring caused by hand-held motion. The distance from the sensor to the surface was measured with a Bosch GLM 50c laser with a range from 0.05 to 50.00 m and an accuracy of ±1.5 mm. To avoid camera vibration and image blurring, a tripod was used during the data acquisition process. The images obtained correspond to cracks located in a bridge column, multiple bridge abutments, and an arch of a bridge. These structures present different levels of smoothness in the concrete and different tones, some being darker than others. This gave a significant variability in the type of concrete surface that was studied. Examples of the different types of concrete surfaces are presented in Figure 1.

In order to validate the results determined using the suggested method, it was necessary to determine the crack widths along different sections of the cracks. For this purpose, a crack magnifier with a scale of 20 mm and divisions of 1/10 mm was employed for direct measurements on the concrete surfaces. For each crack, measurements were performed at multiple sections along the crack, allowing for the collection of multiple width measurements. The positions of these measurement were marked to identify where each measurement has been performed and facilitate their identification it in the images. The images captured correspond to cracks where measurements using the crack magnifier had been performed. In total, 29 cracks were captured using the camera: four images correspond to cracks located in the column of a road bridge, nine images correspond to a concrete wall, two images correspond to cracks in the abutment of a road bridge and fourteen to cracks located in the arch of a road bridge. Within these cracks, a total of 409 sections showcasing different widths were measured using the crack magnifier. These measurements serve as ground truth against which the results of the proposed method are compared.

### 2.2. Pre-Processing

In the initial pre-processing step, the images were divided into sub-images of 224 × 224 pixels. The sub-image approach was used to generate threshold values for each specific section. By doing this, the determined thresholds were specific to each sub-image, avoiding a single threshold for the entire image.

Subsequently, the sub-images were transformed into greyscale, following the standard practice of combining the intensities of the RGB components with NTSC coefficients [33]. The conversion simplifies the analysis of pixel intensity distribution, since each pixel has only one value between 0 and 255. The transformation from RGB to greyscale was carried out using Equation (1) [34]:(1)Y=0.299R+0.587G+0.114B
where *R*, *G*, and *B* correspond to red, green, and blue channel intensities, respectively.

### 2.3. Crack Segmentation Based on the Median Absolute Deviation

This stage of the methodology focused on the greyscale distribution of the pixels. Figure 2 illustrates two regions within an image, one where a sub-image does not contain a crack (window A) and another when it does (window B).

Figure 3 shows the pixel intensity distributions for both sub-image A and B. Figure 3a shows the typical intensity distribution for a sub-image devoid of cracking. The intensities vary but without extreme values that can be considered as intensity discontinuities. However, Figure 3b corresponds to the pixel intensity distribution for sub-image B, where cracking is present. While some intensity variation is evident in certain areas, a clear discontinuity corresponding to the crack becomes apparent. The crack is distinctly identifiable as the region with the lowest values of pixel intensity.

Figure 4 and Figure 5 present both sub-images, as well as their corresponding intensity histogram distributions. The distribution for sub-image A resembles a normal distribution while, for sub-image B, the curve has a tail of lower values in the greyscale, corresponding to pixels belonging to the crack. Since these dark pixels in the tail represent the crack, those are the pixels of interest and are considered as outliers in the algorithm. Outlier detection is typically achieved by determining a threshold, taking the mean, and adding/subtracting the standard deviation a certain number of times. However, this method may cause errors when the distribution is not normal. Furthermore, the mean and the standard deviation are heavily impacted by outliers, making it improper in some cases [35].

To address this issue, this paper used the median as an indicator of the tendency for the greyscale intensity which is less sensitive to outliers [35], and calculated the Median Absolute Deviation (MAD).

The MAD is a scale estimator that measures the variability of a dataset and it can be used for outlier detection. The MAD is calculated as:(2)$$MAD=median\left(\left|X_{i\;}-\tilde{X}\right|\right)$$
where *X_i_* is each element of the series, and $\tilde{X}$ is the median of the series. With this, a threshold *T* can be calculated, as in [35]:(3)$$T=\tilde{X}-k\times MAD$$
where *k* corresponds to a scale factor. Once the threshold is obtained, the image is filtered; if the value of the pixel in the source image is higher than *T*, the pixel in the resulting image is equal to *T*, otherwise, the value of the pixel remains the same.
(4)$$\mathrm{result}\left(x,y\right)=\left\{\begin{array}{c} T\;\;\;\;\;\;\;\;\;\;\;\mathrm{input}\left(x,y\right)\geq T\\ \mathrm{input}\left(x,y\right)\;\;\;\mathrm{input}\left(x,y\right)<T \end{array}\right.$$

For a normally distributed series, where k=3.5, the values within an interval of μ±k×σ, being μ the mean and σ the standard deviation, correspond to 99.7% of the values of the series and, thus, 0.3% of the values will be outliers. A sensitivity analysis is presented for $k=\left[2,\;2.2,\;2.5,\;3,\;3.5\right]$.

The threshold obtained using the MAD depends on the distribution of intensities in each sub-image. Therefore, the threshold value is not the same for the different regions containing a crack but, instead, it is updated for each region. Filtering the sub-images using the threshold *T* provides a crack region from where the edges and centreline of the crack are obtained, and the width is calculated.

### 2.4. Edge Detection

Edges are regions where the intensity changes abruptly and edge detection algorithms use this fact to delimit features present in the images. Gradient-based edge detectors, such as Sobel or Prewitt edge detectors, use derivative filters to detect this change in intensity. Laplacian detectors, on the other hand, use second-order derivative filters. The Laplacian $L\left(x,y\right)$ of an image with pixel intensities $I\left(x,y\right)$ is defined as [27]:(5)$$L\left(x,y\right)=\frac{\partial^{2}I}{\partial x^{2}}+\frac{\partial^{2}I}{\partial y^{2}}$$

The Laplacian method constitutes the convolution mask used to traverse the image and set the edge pixels as having negative intensity values.

### 2.5. Width Calculation

The method proceeded to calculate the width of the crack at each pixel, as part of the centreline. First, the centreline (skeleton) of the crack region was determined with the skeletonize python function from the scikit-image library. At each point of the skeleton, the perpendicular line to the direction of the skeleton was obtained and the nearest edge pixels along the perpendicular line selected (see Figure 6) [28]. The distance between the selected edge pixels provided the width measurement for the corresponding skeleton pixel. The measurements were obtained in terms of pixels and transformed into the actual width using the pinhole camera model to calculate the pixel size $p_{w}$ and following the formula defined as [36]:(6)$$p_{w}=\frac{\tan\frac{\alpha }{2}\times \left(u\right)\times 2}{n_{w}}$$
where $p_{w}\;$ is the pixel size, *u* is the distance from the camera to the surface, α is the lens angle of view, and $n_{w}$ is the resolution of the camera sensor.

With the measurements in terms of millimetres as the predicted value, the error in the width calculation was obtained using the values measured with the crack magnifier as the true values and applying the formula:(7)$$Error\left(\%\right)=\frac{Predicted\;value-True\;value}{True\;value}\times 100$$

### 2.6. Method Performance

The performance evaluation involved different criteria: the Probability of Detection (POD), accuracy, recall, precision, F2 score metrics, and the accuracy of crack width estimation.

The POD is intended to determine the ability of a method to detect a crack of a given size, defining the largest crack width that the method can overlook. Crack width accuracy is determined by comparing the manually measured widths using the crack magnifier with the method’s estimation for each of the points gathered. To further validate the results of the method, the accuracy, recall, F2 scores, and precision metrics were calculated. The number of true positives/negatives and false positives/negatives were recorded manually for each crack and compared with the results obtained. A True Positive (TP) corresponds to a pixel belonging to a crack that has been correctly identified by the method. A False Positive (FP) is a pixel from the background that has been marked as part of a crack by the method. A True Negative (TN) is a pixel belonging to the background that has been correctly identified as such and not marked in the final image. The False Negative (FN) corresponds to a pixel that is part of a crack but that has not been detected as such and, thus, is not marked in the final image. The calculations of the metrics are performed according to Equations (8) to (11), according to [37,38].
(8)$$Accuracy\left(Acc\right)=\frac{TP+TN}{TP+FP+TN+FN}$$
(9)$$Recall\left(Re\right)=\frac{TP}{TP+FN}$$
(10)$$Precision\left(Pr\right)=\frac{TP}{TP+FP}$$
(11)$$F2\;score=\frac{5\times Pr\times Re}{4\times Pr+Re}$$

By carrying out these steps, we evaluated the capability of the method to detect cracks of different widths and determine the accuracy of the width estimations.

Accuracy corresponds to the number of correct predictions made as a ratio of all predictions made. Recall is the measure of how often the method correctly identified positive observations from all the actual positive cases (the true positives + the false negatives). Precision is the measure which determines how often the positive predictions of the method are correct by considering the positive observations (true positives) and all the observations it labelled as positive (the true positives + the false positives). The F2 score is a measure to balance precision and recall; it corresponds to the weighted harmonic mean of the precision and recall. The F2 score gives more weight to recall than to precision since minimizing false negatives is the main concern.

## 3. Results

We applied the MAD method for the different cracked images gathered using the different values of *k*.

Figure 7 presents the different outcomes obtained by applying the method to one of the studied cracks. In each case, the method determined the MAD value for each sub-image and with the corresponding *k* value the threshold *T* was obtained to generate the threshold image with the crack marked in yellow pixels. The width was determined for each crack at each point of the skeleton using the procedure presented in Section 2.5. In each case, the dark pixels treated as outliers by the method have been marked, creating a continuous slender line that forms the crack. This means the method identifies the pixels of interest. For each value of *k*, cracks are highlighted however, the amount of noise presented and the width of the crack change. For a low value of *k* (such as two in Figure 7a), the crack is highlighted in the image and, in the upper part of the image, different elements can be seen around the crack. These elements do not have an elongated shape as cracks usually do. Instead, they appear as dots or rounded elements, constituting noise in the image. As *k* increases, as in Figure 7b–e, the components considered as noise decrease in size and number. This same process is noticeable along the crack. For *k* = 2.5, the noise in the image’s upper and lower parts has decreased considerably concerning Figure 7a with a value of *k* = 2.0. This trend is even more noticeable for *k* = 3.5, where many dot elements have disappeared.

Furthermore, the width of the crack is also affected, particularly at the tip of the two cracks present in the sample. The reduction in width can also be seen as a reduction in the crack area leading to parts of the crack being missed (gaps).

### 3.1. Probability of Detection

The images were annotated manually at the crack locations measured with the optical crack magnifier, to generate the POD curves. In this context, a *hit* refers to the case where a manual marker in the image coincides with the method marking. Whenever the point is not marked by the method, it is considered a *miss*. Based on the *hit/miss* results for each size of crack, the percentage of detected cracks for a given size was calculated. This allows the generation of the Probability of Detection (POD) curves, which illustrate the method’s performance in terms of crack size detection. Following these results, we define the POD as:(12)$$POD=\frac{hit}{hit+miss}$$

The *hit* results for the different crack sizes and different values of *k* are presented in Table 1, where *a* corresponds to the actual crack width, and *n* is the number of measurements performed for the corresponding crack width. For instance, out of the 409 values obtained, 49 correspond to crack sections that are 0.5 mm wide.

Given that different values of *k* are used for the MAD calculation, POD versus crack width curves are generated for each value of *k*. By doing this, we determined the probability of detecting cracks of different sizes for each value of *k*. Figure 8 presents the POD curves for the study. The dashed vertical line represents the 90% confidence rate which, in this case, indicates the point at which all values of *k* have at least a POD = 90%. Similarly, the 95% confidence value is represented for a POD = 95% with a dashed-dotted line. The 90% and 95% detection rates for each value of *k* are given in Table 2. The 50% detection rate is not displayed, since the lowest POD for the different cases is 64%. The lowest *a90* and *a95* were obtained for the case in which *k* was 3.5, with values of 0.19 and 0.21 mm, respectively.

From the POD results, it can be seen that, by using the MAD segmentation approach presented in this paper, the largest crack width that is possible to miss is 0.19 mm, when using a *k* value of 3.5. By using a *k* value of 2.5, the probability of detecting cracks that have a width of 0.1 mm is 97%. Among the different values used for *k*, 2.0, 2.2, and 2.5 have the lowest values for a*90* and a*95*, showing that the method is efficient for detecting crack widths of 0.1 mm and greater. This can be a useful tool for inspectors, since they can obtain the small crack regions that can sometimes be overlooked when visually reviewing images.

### 3.2. Width Calculation

The width calculations were performed for the different values of *k* at each pixel of the corresponding skeleton, creating width calculations along the crack. The points where the crack magnifier was used to determine the actual values of width were compared to the estimations made by the method. The error was calculated for each case and the mean values were determined for each crack size and each *k* value; these are presented in Table 3. The method detects the regions that correspond to the crack where the width is 0.1 mm but, in general, it overestimates the width by more than double (>100%). The error is similar for the different values of *k*, although decreasing. Indeed, the mean error tends to decrease with higher values of *k* and higher values of crack width. Although in the majority of cases there is an overestimation of the crack width, using *k* = 3.5 generates negative values corresponding to an underestimation of the width.

Errors are more pronounced for smaller crack widths. One of the contributing factors is that, in order to identify regions corresponding to crack widths of 0.1 mm, the *k* value for the MAD is set to lower values. This implies that the threshold value is closer to the median, resulting in a larger number of intensities which can be categorised as outliers. As a result, pixels in the neighbourhood of a crack, which have close intensity values to the pixels within the crack, end up being marked as part of the crack, effectively enlarging the crack area. The selection of the *k* influences the width estimation as well as the probability of detection, as presented previously. For lower values of *k*, the POD is higher since there are no gaps in the continuity of the crack, but the noise and the overestimation of the width increase. For higher values of *k*, gaps might appear, but the width estimation is closer to reality. Considering these two scenarios, the selection of *k* becomes a trade-off between the amount of noise accepted, the need to detect the full extension of the crack, and the width of the crack.

Additionally, it is important to consider that the real size of the pixels corresponds to the minimum value for the target crack size. This means that an error of one pixel for a crack measuring 0.1 mm corresponds to an error of 100%. In order to mitigate this, it would be necessary to have a pixel size smaller than 0.1 mm, either by using a camera with higher resolution or by setting the camera closer to the concrete surface.

### 3.3. Comparison of Methods

This section compares two established methods: the Otsu method and the Niblack method. These methods correspond to some of the most common methods used for image thresholding and are found in guide textbooks [39]. Furthermore, these methods are used by other researchers to detect cracks, making them relevant methods for comparison [8,24,40,41]. Figure 9 presents two examples of the results obtained for the different methods.

In Figure 9a,e, we present two unprocessed images for different types of concrete containing cracks. Figure 9b,f present the results obtained using the suggested method with a *k* value of 2.5. Subsequently, Figure 9c,g show the results of applying the Otsu method to both images. Finally, Figure 9d,h show the results obtained after applying the Niblack method.

In the case of the Otsu method, the crack cannot be seen in a clear way, while the MAD and Niblack methods clearly highlight the cracks. However, in these two cases, some noise is represented by areas incorrectly highlighted as cracks. Despite the crack being distinguishable, the noise limits the possibility of an inspector or responsible person being able to focus solely on the damage. Indeed, when using the Otsu method, the noise makes it difficult to clearly delineate the crack and determine the extent of the damage.

On the other hand, for the Niblack method, the crack is delineated when the background surface is homogeneous, to some extent. In the case of the second image, variations in the concrete colour and different sizes of aggregate make the surface more heterogeneous; this situation generates noise that can make the crack more difficult to distinguish in some regions. This case would require a higher level of attention for an inspector to be able to notice the limits of the crack effectively. 

When using the MAD method, the cracks are clearly delimited, simplifying their assessment for an inspector. While the method generates noise similar to the previous cases, the quantity of noisy regions is considerably less than for the Otsu or Niblack methods.

From the considered images, the manually marked regions were used to determine the number of True/False Positive/Negative detections for comparison of the methods. The results of these metrics are presented in Table 4.

The accuracy of the MAD method demonstrates superior results when compared with the Otsu and Niblack methods, with the lowest value being 97%. Recall represents the number of observations the method correctly identified as cracks from all the possible positive cases. For this metric, the Otsu method performed best but with a precision of only 5%. Precision corresponds to the rate of observations the method correctly identified as cracks, relative to all the observations classified as cracks. Although Otsu has the best results of recall, the MAD performed better for that metric than Niblack, while achieving the best results for precision.

In this context, a high level of recall is important as it indicates how effectively the method detects relevant information (cracks in this case). Precision gives a measure of false alerts, which can be considered less critical than correctly identifying damage, making recall more significant. Taking this into account, the MAD method achieves the best results for the F2 score, which relates recall and precision and puts more emphasis on recall.

## 4. Conclusions

The method applies the MAD to determine a threshold value for different sub-images of an image where there is a crack. The calculation of the MAD for each sub-image allows us to obtain an adaptable threshold value, adjusted to the characteristics of each particular sub-image, rather than using a general threshold for the entire image. This approach has proved to be efficient for determining cracks in the images; detecting cracks with widths of 0.1 mm at a probability of 97%, when using low values of *k*, and 64% with the highest value of *k*. The method delivers results with minimum pre-processing requirements, other than converting the image into greyscale.

The results indicate that lower *k* values (2.5 and below) give superior results in terms of crack detection, but higher errors in their quantification. On the contrary, the highest value of *k* gives suboptimal detection results, but the best in terms of quantification. To enhance the method’s performance, a combination of two *k* values can be explored, one prioritising the crack detection and another prioritising the quantification, creating a balance between the two aspects.

It is important to acknowledge a limitation for the method, since it tends to overestimate the crack width in most cases. The overestimation is attributed to the *k* values selected, which have a tendency to prioritise the detection of crack areas by applying a larger threshold value. Additionally, the error is influenced by the size of the pixel, since an error of one pixel can represent a 100% error, in terms of the real value. Despite this limitation, the overestimation can be perceived as a conservative measure, assuring that the method is overestimating damage and, thus, a harmful situation for the structure.

Likewise, the presence of noise in the results should be considered. However, when considering the practical application of this method and the evaluation of the results by an inspector, the human perspective plays a crucial role in distinguishing the noise from vital information. As presented in the comparison of the methods, the MAD method outputs the lowest values for noise in comparison to the Otsu and Niblack methods, showing the highest values of precision and F2 score. The method highlights regions containing cracks, allowing an inspector to focus on those specific areas without having to examine the entire picture. This reduces the possibility of the inspector missing a crack due to fatigue, lack of concentration, or having cracks blending in with the background. Although part of the noise could be wrongly identified as a crack, this conservative approach is preferred over missing regions with cracks, thus making the results more reliable.

In future studies, hybrid processing will be implemented, integrating various values of *k* to balance between prioritizing crack detection and width estimation. Additionally, a post-processing step will be introduced after applying the MAD procedure to eliminate noise based on the shape of detected elements. This involves analysing the relationship between length and width to identify and remove round elements that correspond to noise, thereby enhancing the precision of crack identification.

## Figures and Tables

**Figure 1 sensors-24-02736-f001:**
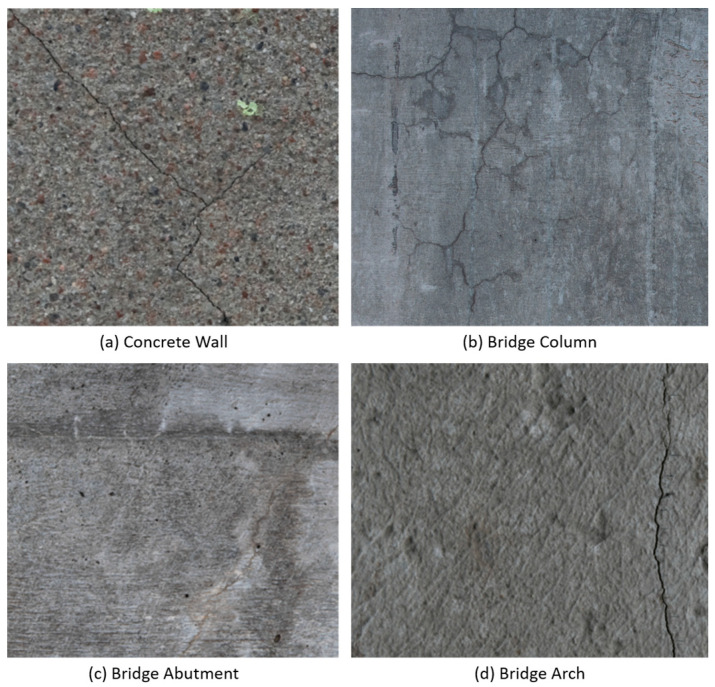
Surfaces studied: (**a**) Concrete Wall, (**b**) Bridge Column, (**c**) Bridge Abutment, (**d**) Bridge Arch.

**Figure 2 sensors-24-02736-f002:**
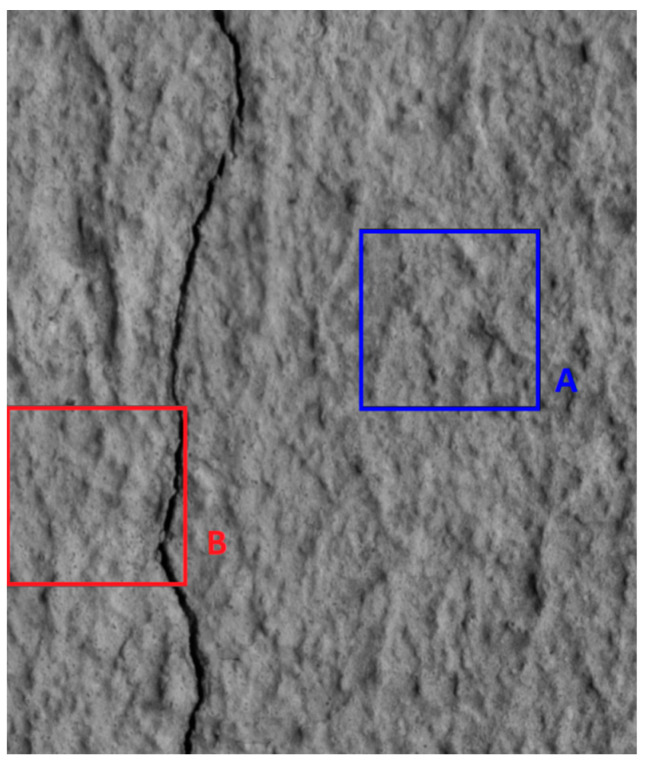
Non-Cracked region (Window A) and Cracked region (Window B) on image.

**Figure 3 sensors-24-02736-f003:**
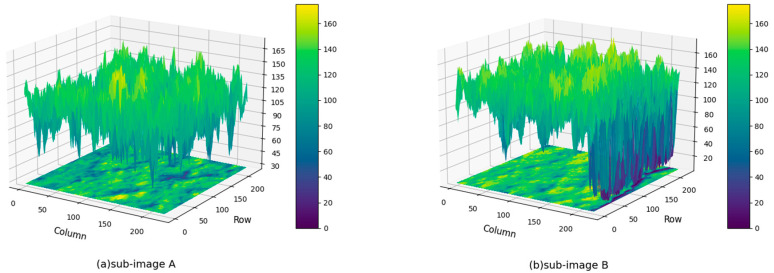
Three-dimensional greyscale distribution: (**a**) sub-image A, (**b**) sub-image B.

**Figure 4 sensors-24-02736-f004:**
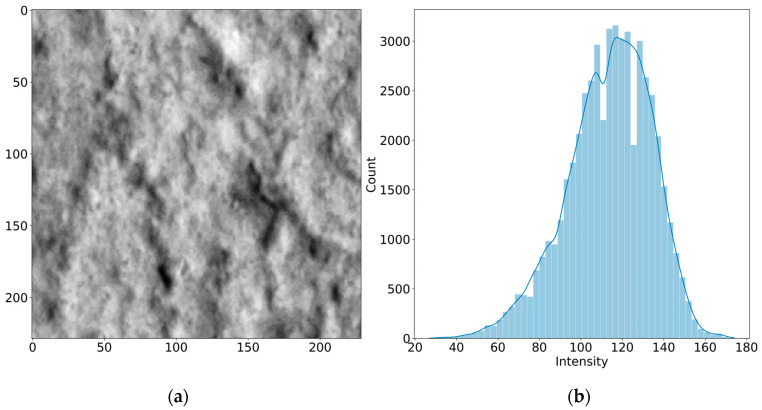
Uncracked concrete section: (**a**) sub-image A, (**b**) intensity histogram distribution.

**Figure 5 sensors-24-02736-f005:**
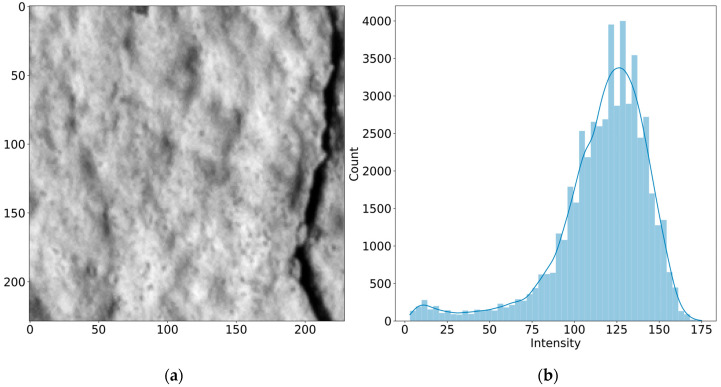
Cracked concrete section: (**a**) sub-image B, (**b**) intensity histogram distribution.

**Figure 6 sensors-24-02736-f006:**
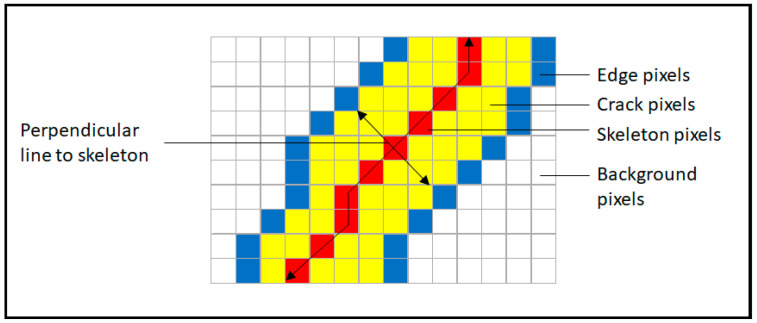
Crack width calculation.

**Figure 7 sensors-24-02736-f007:**
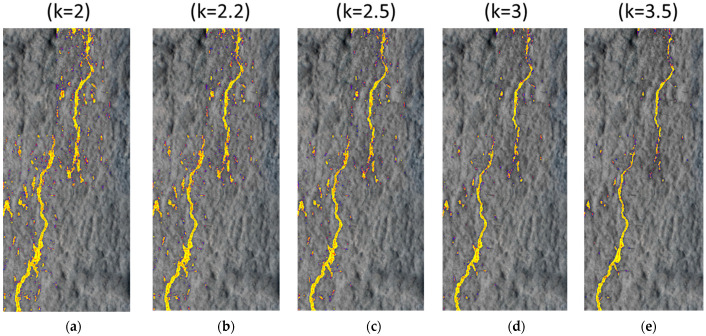
Crack results for different values of *k*: (**a**) *k* = 2, (**b**) *k* = 2.2, (**c**) *k* = 2.5, (**d**) *k* = 3, (**e**) *k* = 3.5.

**Figure 8 sensors-24-02736-f008:**
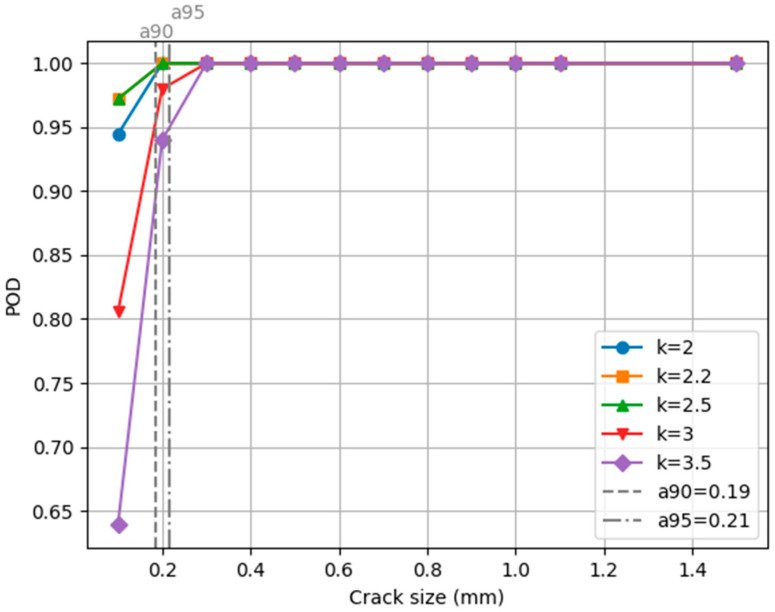
POD curves for different values of *k* for the different crack sizes studied.

**Figure 9 sensors-24-02736-f009:**
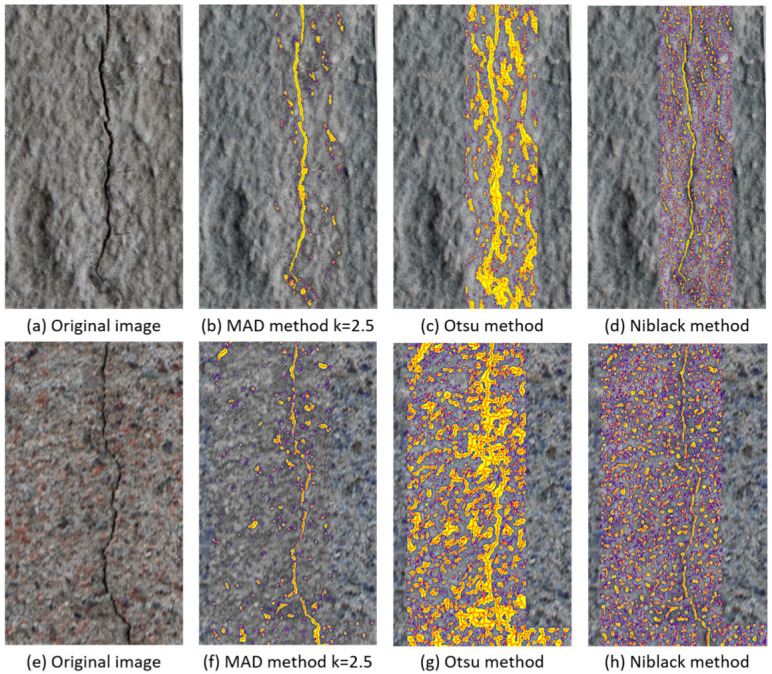
Comparison of MAD, Otsu, and Niblack methods.

**Table 1 sensors-24-02736-t001:** Number of hits for the different crack sizes using different *k* values.

Crack Width	Number of Measurements	Number of Hits for Different *k* Values				
*a* (mm)	*n*	*k* = 2.0	*k* = 2.2	*k* = 2.5	*k* = 3.0	*k* = 3.5
0.1	36	34	35	35	29	23
0.2	50	50	50	50	49	47
0.3	31	31	31	31	31	31
0.4	27	27	27	27	27	27
0.5	49	49	49	49	49	49
0.6	25	25	25	25	25	25
0.7	32	32	32	32	32	32
0.8	38	38	38	38	38	38
0.9	20	20	20	20	20	20
1.0	43	43	43	43	43	43
1.1	13	13	13	13	13	13
1.5	9	9	9	9	9	9

**Table 2 sensors-24-02736-t002:** Detection rates *a90* and *a95* for different values of *k*.

*k* Value	*a*_90_ (mm)	*a*_95_ (mm)
2.0	0.10	0.10
2.2	0.10	0.10
2.5	0.10	0.10
3.0	0.15	0.18
3.5	0.19	0.21

**Table 3 sensors-24-02736-t003:** Crack width results.

Crack Width (mm)	Crack Width (Pixels)	Mean Error (%)				
		*k* = 2.0	*k* = 2.2	*k* = 2.5	*k* = 3.0	*k* = 3.5
0.1	1	165	161	193	140	120
0.2	2	151	101	67	27	81
0.3	3	85	71	36	32	19
0.4	4	87	78	35	41	16
0.5	5	66	50	35	23	13
0.6	6	52	48	34	32	17
0.7	7	47	34	53	53	1
0.8	8	36	29	37	15	−1
0.9	9	74	17	35	14	0
1.0	10	44	35	91	2	15
1.1	11	21	17	41	63	54
1.5	15	7	9	27	−11	−19

**Table 4 sensors-24-02736-t004:** Comparison of validation results.

Method		Accuracy	Recall	Precision	F2 Score
	*k* = 2.0	97%	93%	20%	32%
	*k* = 2.2	98%	90%	24%	38%
MAD	*k* = 2.5	98%	86%	31%	44%
	*k* = 3.0	99%	75%	41%	52%
	*k* = 3.5	99%	63%	51%	54%
Otsu		86%	99%	5%	10%
Niblack		91%	88%	7%	12%

## Data Availability

The data used in this study are available on request from the corresponding author.

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
