# Peer review of "Image-Based Concrete Crack Detection Method Using the Median Absolute Deviation"

_sensors, 2024, doi:10.3390/s24092736_

Round 1
Reviewer 1 Report
Comments and Suggestions for Authors
This paper proposes an adaptive threshold method based on Median Absolute Deviation (MAD) in images of concrete with cracks for detecting and quantifying them. Results by the proposed method are compared to those from traditional methods such as the Otsu method and Niblack method, and show better performance in several selected metrics. The manuscript is well written and well organized. Therefore, the reviewer supports the publication of this paper in Sensors, if the following points would be considered in the final version.
- Line 194, how do authors determine the number of sub-images?
- Line 282, although authors show these metrics in following equations, it would be helpful to explicitly explain them.
- Conclusions, it would be helpful to add some comments on future work based on limits metioned in this section.
- Overall, it would be helpful to provide reasons for selecting Otsu and Niblack methods instead of other methods as references.
Reviewer 2 Report
Comments and Suggestions for Authors
This article proposes a new method for detecting and quantifying concrete cracks, and accurately identifies the width of cracks. From the perspective of content and methods, this article has strong innovation and can be published with minor modifications:
1. More relevant information about the dataset needs to be supplemented, as the quality of the dataset directly determines the recognition accuracy.
2. Which part of the bridge does the crack data belong to?
3. Can the dataset be made public to facilitate readers and relevant researchers in reproducing crack identification.
4. In the introduction section, it is necessary to add more information about the causes of bridge cracks, such as:
Steel and Composite Structures, 2024, 50 (3): 363-374
Structures, 2023, 57: 104996
5.In Figure 7, the crack recognition effect caused by different k values feels similar, without any detailed differences. It is suggested to add more discussion on k values.
6.What is the probability of crack identification in section 3.1? Is it derived based on Bayesian theory?In terms of Bayesian analysis, the following references can be added:
Probabilistic Engineering Mechanics, 2023, 73:103475&103483.
7.How is the width determined? Usually, the width in an image is pixel wide. How can it correspond to the actual width?
8.The discusion section needs further elaboration on how to identify cracks and their widths, and how different thresholds affect accuracy.
9.The conclusion section needs to identify the shortcomings of this article and future research directions.
